# Saponin Improves Recovery of Bacteria from Orthopaedic Implants for Enhanced Diagnosis Ex Vivo

**DOI:** 10.3390/microorganisms13040836

**Published:** 2025-04-07

**Authors:** Tiziano Angelo Schweizer, Adrian Egli, Philipp P. Bosshard, Yvonne Achermann

**Affiliations:** 1Department of Dermatology, University Hospital Zurich, University of Zurich, 8091 Zurich, Switzerland; 2Department of Cranio-Maxillo-Facial and Oral Surgery, University Hospital Zurich, University of Zurich, 8091 Zurich, Switzerland; 3Institute of Medical Microbiology, University of Zurich, 8006 Zurich, Switzerland; 4Internal Medicine, Hospital Zollikerberg, 8125 Zollikerberg, Switzerland

**Keywords:** skin antisepsis, photodynamic therapy (PDT), daylight, 5-aminolevulinic acid, methyl-aminolevulinate, bacteria

## Abstract

Biofilm formation on orthopedic joint implants complicates diagnosis of periprosthetic joint infections (PJIs). Sonication of explanted orthopedic implants for diagnostic enhances pathogen detection, but it shows limitations in sensitivity and handling. We investigated whether the biosurfactant saponin could improve bacterial recovery from orthopaedic implants and thereby enhance infection diagnosis ex vivo. Orthopaedic material discs of 1 cm diameter were contaminated with different clinical bacterial PJI isolates. Biofilms of *Staphylococcus epidermidis*, *Staphylococcus aureus*, *Escherichia coli*, *Cutibacterium avidum*, and *Cutibacterium acnes* were grown on the discs, which were then treated with either saline solution or various concentrations of saponin. Next, the discs were vortexed or sonicated. Colony-forming units (CFUs) enumeration and time-to-positivity of liquid cultures were determined. Additionally, a novel 3D PJI soft tissue in vitro model was established to validate these findings in a more representative scenario. Median CFU enumeration showed that 0.001% (*w*/*v*) saponin as compared to saline solution increased CFUs recovery by 2.2 log_10_ for *S. epidermidis*, 0.6 log_10_ for *S. aureus*, 0.6 log_10_ for *C. avidum*, 1.1 log_10_ for *C. acnes*, and 0.01 log_10_ for *E. coli*. Furthermore, saponin treatment resulted in a >1 log_10_ increase in *S. epidermidis* CFU recovery from implants in the 3D tissue model compared to standard saline sonication. With that, we propose a novel two-component kit, consisting of a saponin solution and a specialized transportation box, for the efficient collection, transportation, and processing of potentially infected implants. Our data suggest that biosurfactants can enhance bacterial recovery from artificially contaminated orthopedic implants, potentially improving the diagnosis of PJIs.

## 1. Introduction

Periprosthetic joint infections (PJIs) are a serious complication following the implantation of an orthopedic device. Despite enhanced efforts to prevent infections, the incidence of infections in hip and knee arthroplasties is rising. PJIs affect up to 1–2% of patients with such a procedure [1], even with standard preoperative skin antisepsis and perioperative antibiotic prophylaxis [2]. PJIs are associated with increased morbidity, prolonged antibiotic therapy, and repeated surgical interventions. Furthermore, the cost of treating a single episode of PJI exceeds $45,000 [1].

The most frequently isolated microorganisms in PJIs include staphylococci, streptococci, and anaerobic bacteria such as *Cutibacterium acnes* and *Cutibacterium avidum* [1]. These bacteria are known to form biofilms, which are a major factor in the challenge to treat PJIs [3]. Despite the many therapeutic challenges, the diagnostic of a PJI is also not straightforward for most microbiological laboratories. Often, patients were pre-treated with antibiotics, which result in slow or no growth. Some bacterial species require specific growth conditions, such as *C. acnes*. In addition, intraoperative tissue cultures often show low sensitivity and specificity for pathogen identification [4,5,6], requiring multiple positive samples to confirm infection [7]. To improve pathogen detection, sonication of explanted implants, which dislodges bacterial biofilms, has been used for over a decade [8,9,10]. While sonication generally provides better sensitivity over conventional tissue culture [11,12], this method still has its limitations. First, there are no standardized protocols or equipment for sonication [10,11,12,13], making it a laboratory-developed test in most cases and resulting in variability when comparing studies. Second, sonication fails to achieve high sensitivity; partly because bacteria are less affected by ultrasound [14,15]. Third, detached biofilm bacteria can persist as aggregates in the sonication solution [16,17]. This results in reduced colony-forming unit (CFU) counts, potentially missing the cut-off for diagnosing PJI based on CFUs [18]. As a consequence of these diagnostic challenges, the long time-to-positivity results (TTP) of bacteria in biofilm aggregates results in delayed tailored antibiotic treatments, which affect patient outcomes. Recently, it has been shown that chemical or enzymatic agents have been explored to detach biofilms from orthopedic implants and improve bacterial recovery [13,19]. Dithiothreitol (DTT) has shown promise in vitro, demonstrating increased sensitivity as compared to saline sonication [20], and, in initial clinical trials, showing superior sensitivity over standard tissue samples [21,22]. However, its limited stability, potential toxicity profile, and equal or inferior sensitivity as compared to standard saline sonication in subsequent clinical trials have restricted its clinical application [23,24,25].

Surfactants, which can reduce surface tension and promote detachment of biofilm, might offer a solution [26,27]. However, most surfactants also have cell-permeabilizing effects, limiting their applicability [28,29]. The biosurfactant saponin, derived from *Quillaja* species is notable for its soap-like properties, potential for biofilm detachment, and low toxicity profile [30,31,32]. While enzymes have been extensively studied for biofilm detachment, their application is often limited by poor stability and stringent storage requirements. In contrast, saponin offers optimal stability, even at room temperature [33]. Additionally, as a biological derived compound, saponin poses minimal environmental risk, further supporting its potential for a broader application [34].

As mentioned, sonication lacks standardized equipment and protocols. The devices used for implant processing—simple sterile plastic boxes—require complex and manual manipulations, increasing the risk of contamination and false positive results [9]. Variable volumes of sonication solution further complicate a PJI diagnosis based on CFU cut-off values [18] as there is no correction to the volume of fluid used.

To address these challenges, we investigated whether saponin could enhance bacterial CFUs recovery from biofilm grown on orthopedic material in vitro. We used saponin in combination with sonication or alone. We also explored if this procedure reduced the time-to-results in a 3D soft tissue PJI model.

## 2. Materials and Methods

### 2.1. Bacterial Strains

The following clinical isolates from patients suffering from PJI were used for PDT experiments: *Staphylococcus epidermidis* BCI112, BCI135, and BCI195, *Staphylococcus aureus* BCI175 and BCI187, *Cutibacterium avidum* BCI100, *Cutibacterium acnes* BCI104 and BCI105 and *Escherichia coli* 1 and 2. The bacterial biobank was approved by the institutional review board in Zurich, Switzerland (KEK Nr 2016-00145, KEK Nr 2017-01458). Fresh cultures were prepared from frozen cultures and were put on Brain Heart Infusion (BHI, Becton Dickinson, Heidelberg, Germany) agar plates at 37 °C for 24 h under aerobic conditions for *S. aureus*, *S. epidermidis,* and *E. coli* or under anaerobic conditions for 3 days in the case of *C. acnes* and *C. avidum* (GENbags, bioMérieux, Mary-l’Etoile, France). Liquid cultures were made with colonies from the agar plates in BHI broth in a 37 °C incubator with shaking overnight under aerobic conditions for *S. aureus*, *S. epidermidis* as well as *E. coli* and shaking for three days under anaerobic conditions for *C. acnes* and *C. avidum*. Then, 1:1000 dilutions were made in a fresh BHI medium. Samples were vortexed and used as start inoculum for biofilm assays right away.

### 2.2. In Vitro PJI Biofilm

Biofilms were grown as described previously with minor adaptations [35]. In brief, bacterial liquid cultures in RPMI-1640 + 5% fetal calf serum (FCS, Gibco, ThermoFisher Scientific, Waltham, MA, USA) were used to inoculate sterilized polyethylene (PE), titanium alloy Ti-6AI-4V (TAV), cobalt-chromium-molybdenum (CCM), all provided by all provided by Johnson & Johnson Family of Companies (Zug, Switzerland), and polymethyl methacrylate (PMMA)-based bone cement (PMMA), provided by Heraeus Group (Hanau, Germany) in discs of 1 cm diameter in 24-well plates (Appendix A). The plates were incubated for 3 days at 37 °C for *S. epidermidis*, *S. aureus*, and *E. coli* and for 4 days at 37 °C under anaerobic conditions for *C. acnes* and *C. avidum*.

### 2.3. Biofilm Detachment

After the incubation period, the supernatant was removed and the discs containing the biofilms were transferred to fresh wells. The discs were then washed 3 times with phosphate-buffered saline (PBS, Sigma Aldrich, St. Louis, MO, USA) to remove planktonic bacteria and transferred to 15 mL tubes containing 2 mL of either PBS alone or PBS containing saponin from *Quillaja* sp. (Sigma Aldrich, St. Louis, MO, USA) at various concentrations The tubes with the discs were then assigned to either the Vortex or the Sonication group. The Vortex group tubes were vortexed for 1 min, while the Sonication group tubes were vortexed for 30 s, followed by sonication for 1 min at 40 Hz and a second round of vortexing for 30 s (Appendix A). The resulting vortex- or sonication-fluids were then serially 10-fold diluted and drop-plated on agar BHI plates to enumerate CFUs as well as diluted 1:10 in a fresh BHI medium to visually determine time-to-positivity after 24, 48, 72, and 96 h. Agar plates were incubated at 37 °C for 1 day for *S. epidermidis*, *S. aureus,* and *E. coli* and for 3 days at 37 °C under anaerobic conditions for *C. acnes* and *C. avidum*. Experiments were carried out in duplicate and repeated three times.

### 2.4. Novel In Vitro 3D PJI Soft Tissue Model

To build the model, 24-wells were covered with a 0.5 mg/mL collagen type I solution (Rat Tail, Sigma Aldrich, St. Louis, MO, USA) in DMEM/F12 (Gibco, ThermoFisher Scientific, Waltham, MA, USA) and allowed to solidify for 20 min at 37 °C. Then, a TAV disc was placed in the center of the well. Next, a fibroblast containing collagen solution was prepared. To do so, the human skin fibroblast cell line BJ (ATCC, Manassas, VA, USA) was grown in DMEM/F12 + 5% FCS until confluent and then harvested. A 2 mg/mL collagen solution containing 5 × 10^4^ fibroblasts per 300 µL was prepared in DMEM/F12 + 5% FCS and added to the well plate containing the implant. After an incubation period of 20 min at 37 °C + 5% CO_2_, 200 µL DMEM/F12 + 5% FCS were added to each well. The models were incubated for 3 days at 37 °C + 5% CO_2_. The medium was replaced on day 4 and models were allowed to grow for another 2 days until infection.

### 2.5. Infection and Processing of the Novel 3D PJI Soft Tissue Model

Bacterial overnight cultures in BHI (*S. epidermidis*) were centrifuged and resuspended in PBS. They were then set to an OD 0.5 in PBS and diluted 1:10^6^ in RPMI + 5% FCS to reach an inoculum between 50 and 200 CFUs. Models were then infected with 50 µL of the prepared bacterial suspensions and topped with 150 µL RPMI + 5% FCS. Models were incubated for 3 days at 37 °C + 5% CO_2_. Prior to processing, the supernatant was removed. The implant discs were then separated from the tissue and placed in 15 mL canonical tubes, while the tissues were placed in 2 mL tubes. To each sample, either 2 mL of PBS or saponin 0.001% were added. PBS tissue samples were homogenized in a bead beater with sterile metal beads for 10 min at 30 kHz, while the PBS implants were sonicated for 1 min. Saponin-treated discs and tissues were vortexed for 1 min. Resulting solutions were then 10-fold diluted in PBS and drop-plated on BHI agar plates. Six independent models (one for microscopy, five for CFU quantification), in three independent experimental runs were used.

### 2.6. Confocal Laser Scanning Microscopy

In order to visualize *S. epidermidis* biofilm aggregates on the TAV discs, we imaged biofilms surrounded by human cells and host tissue (Collagen) and single S. epidermidis colonies within the host tissue matrix colonized by human fibroblasts. Models were fixed with 4% PFA for 20 min and then stored at 4 °C until use. They were stained with picrosirius red (PSR) staining solution [0.5 g of Direct Red 80 (Sigma Aldrich, St. Louis, MO, USA) in 500 mL of saturated picric acid (Sigma Aldrich, St. Louis, MO, USA) as described previously [36]. Next, they were washed three times with 0.5% acetic acid prior to one wash with PBS. Then, they were stained with 1:1000 dilution of both Wheat Germ Agglutinin Alexa Fluor^TM^ 488 (ThermoFisher Scientific, Waltham, MA, USA) and Hoechst 33342 (ThermoFisher Scientific, MA, USA) in PBS for 15 min. Samples were visualized and acquired by confocal laser scanning microscopy (CLSM) with a Leica TCS SP8 inverted microscope (Leica, Hessen, Germany) under a 63×/1.4 NA oil immersion objective. The obtained images were processed to 3D images using Imaris 9.2.0 (Bitplane, Schlieren, Switzerland).

### 2.7. Moleculight i:X^TM^ Imaging

One day prior to harvesting, models were supplemented with 5 mM hexyl-aminolevulinate and incubated for 24 h. The supernatant was removed, models washed once gently with PBS and pictures were taken with the Moleculight i:*X*^TM^ (MolecuLight Inc., Toronto, ON, Canada) device according to the manufacturer’s instructions. The Moleculight i:*X*^TM^ device shows image autofluorescence produced by collagenous tissue and bacterial biofilms [37,38,39].

Images were processed with ImageJ v. 1.54.

### 2.8. Statistical Analysis

Data visualization and quantitative analyses were performed using GraphPad Prism 9.2.0 (GraphPad Software, San Diego, CA, USA).

## 3. Results

### 3.1. Finding the Optimum Saponin Concentration for Bacterial Recovery from Biofilms on Implant Discs

To assess the effect of saponin on bacterial recovery, implant discs were colonized with *S. epidermidis* for three days in a physiological medium, resulting in the formation of biofilm-like aggregates on the implant disc surface, representing low-grade infections (Figure 1A). Sonication with saline solution (PBS) resulted in a >1 log_10_ increase in CFU recovery compared to vortexing with saline solution (PBS) (Figure 1B). Increasing the sonication time from 1 to 5 min did not significantly enhance bacterial yield (Appendix A). We then performed a concentration titration of saponin to determine its effect on bacterial recovery from PE implant discs. Vortexing with low concentrations had either no to little effect (at 10^−7^ and 10^−6^% (*w*/*v*) saponin) or increased the CFU recovery to a level comparable to sonication (at 10^−5^ and 10^−4^% (*w*/*v*) saponin). However, concentration ≥0.001% (*w*/*v*) saponin resulted in a remarkable increase of >2 log_10_ CFUs recovery with the highest recovery at 0.001% (*w*/*v*) and a decreasing trend at higher concentrations. No additional benefit was observed when sonication was conducted in PBS containing saponin (Appendix A); therefore, we chose vortexing with 0.001% (*w*/*v*) saponin for all subsequent experiments. Saponin vortexing also resulted in >2 log_10_ CFU recovery of bacteria from TAV, CCM, and bone cement PMMA discs compared to saline sonication (Figure 1C).

### 3.2. Saponin Effectively Reduces Time-to-Positivity (TTP)

We then assessed the liquid culture time-to-positivity (TTP) for low-grade infections. Solutions obtained from either saline sonication or saponin vortexing of *S. epidermidis* biofilms described above were diluted 1:10 and incubated in a fresh liquid medium to monitor TTP. All saponin-treated samples showed 100% culture growth after 48 h, while saline sonication samples required four days to reach 100% culture growth (Figure 1D).

### 3.3. Saponin Enhances Recovery of the Most Prevalent PJI-Causing Bacteria from Orthopedic Material

Since strain-dependent effects might occur, we chose to assess the effect of 0.001% (*w*/*v*) saponin vortexing as compared to saline sonication for further clinical PJI isolates, representing the major causative species of PJI. For *S. epidermidis* (total 3 PJI isolates), a median 2.2 log_10_ CFUs increase was observed with saponin, while for *S. aureus* (2 PJI isolates), a 0.6 log_10_ CFU increase was observed with saponin (Figure 2). Concerning *Cutibacteria*, we observed a median 0.6 log_10_ CFUs increase for *C. avidum* (2 PJI isolates and a median 1.1 log_10_ CFUs increase for *C. acnes* (2 PJI isolates) (Figure 2). For *E. coli* (2 PJI isolates), we only observed a negligible median 0.01 log_10_ CFUs increase (Figure 2).

### 3.4. Saponin Enhances Recovery of S. epidermidis from Orthopedic Implant Material in a Novel 3D PJI Soft Tissue Model

To evaluate whether saponin would maintain its effect on bacterial recovery in a more physiological context, we developed a simple 3D model. Infection of the model with *S. epidermidis* for three days resulted in the formation of small bacterial biofilm aggregates on the implant disc in contact with host tissue and cells (Figure 3A, left “implant”) as well as single colonies distributed throughout the tissue surrounding the implant (Figure 3A, right “tissue”). Macroscopic evidence of bacterial infection and biofilm formation was also observed using the Moleculight i:*X*^TM^ device (Figure 3B), which exploits bacterial autofluorescence (red) caused by porphyrin production [40]; we observed a significant increase in red signal over green (collagen) signal, for infected models as compared to uninfected models. Upon separating the implant disc from the cell layers, bacterial biofilm formation was primarily evident at the tissue-implant interface as well as on the implant itself. Tissue and implant were separated and then treated either with saline sonication or with saponin vortexing. Saponin-vortex-treated implants showed significantly higher CFU recoveries compared to saline sonicated implants, while no difference was observed in tissue samples (Figure 3C).

## 4. Discussion

Biofilm formation on orthopedic joint implants is a significant challenge to identify the causative pathogens of PJI. Current guidelines recommend ≥2 culture positive samples (i.e., tissue, fluid, etc.) with the same microorganism or ≥50 CFUs/mL of any organism from sonication. Sonication of explanted implants has significantly improved the sensitivity of microbiologic diagnostics for PJIs by dislodging bacteria from biofilms adhered to prosthetic surfaces [8,10,13,18]. However, on the one hand, bacterial recovery from biofilms is still suboptimal and, on the other hand, the currently used sonication process with simple plastic boxes requires multiple manipulations that increase the risk of contamination and false positive results. Therefore, we investigated whether surfactants, specifically the biosurfactant saponin, could increase bacterial recovery from orthopedic material in vitro. We demonstrated that saponin strongly enhances recovery from bacteria causing PJI particularly *S. epidermidis*, which is often difficult to diagnose. Importantly, saponin can be used in combination with or independent of sonication, resulting in significantly higher recovery as compared to standard saline sonication. To address contamination issues, we propose a novel two-component kit to collect, transport, and process potentially infected implants with a saponin solution in an efficient and sterile manner.

We found that saponin’s effect was concentration dependent. Vortexing with 0.001% (*w*/*v*) saponin was significantly more effective than saline sonication. While concentrations >0.001% (*w*/*v*) led to a trend of decreasing CFU recovery, it still was higher than with sonication, even at 10% (*w*/*v*) saponin. Saponin is known to be potentially toxic towards prokaryotic and eukaryotic cells [32]; however, no toxicity against bacteria was observed at 0.001% in our experiments. While this concentration may be toxic to human cells, it is not a concern, as the implants are treated with saponin after being retrieved from the body. This is not the first study showing increased CFU recovery from implants with a substance compared to standard sonication. DTT, for instance, has been extensively studied and has demonstrated superiority in both in vitro and early clinical studies for CFU recovery and sensitivity [13,21]. In the study by Drago et al. [20], DTT achieved a maximum increase of 0.5 log_10_ in CFU recovery for both *S. epidermidis* and *S. aureus* on PE and titanium discs. In contrast, our study showed that saponin led to an increase of more than 2 log_10_ for *S. epidermidis* and 0.5 log_10_ for *S. aureus*. However, a direct comparison of DTT and saponin across these studies may be biased due to differences in experimental settings. Moreover, follow-up clinical studies have shown that DTT has inferior sensitivity compared to sonication, possibly due to stability issues of DTT [23,24]. For this reason, we opted not to include DTT in our experimental setting. Enzymes and other substances may also face similar stability and cost challenges [34], whereas saponin is known for its excellent stability in solution, even at room temperature [33].

Among the tested pathogens causing PJIs, saponin vortexing demonstrated the highest efficacy in recovering *S. epidermidis* from implants. This is an important finding since *S. epidermidis* is a major cause of PJIs [1] in particularly chronic PJIs. Chronic PJIs are more challenging than acute infections because they often present with only pain, making them harder to identify and treat [40,41,42]. In sonication studies, *S. epidermidis* is one of the bacteria known to withstand dislodgment from surfaces [11], which may explain some of the difficulties in diagnosing *S. epidermidis* PJIs. Furthermore, we chose an incubation time that would promote the formation of biofilm clusters of *S. epidermidis*, which also results in relatively low CFU recovery by sonication, a result similar to what is typically observed with clinical orthopedic implants [43].

For all other tested bacteria (*S. aureus*, *C. avidum*, *C. acnes*, and *E. coli)*, we observed an effect of saponin vortexing in enhancing bacterial recovery from implant surfaces, except for *E. coli*. *E. coli* already showed relatively high recovery with sonication compared to the other isolates. Although the exact reason for the lack of increased recovery with saponin is unclear, it is possible that the attachment mode and biofilm matrix composition of *E. coli* [44,45] make it more susceptible to sonication than other bacteria. The increased recovery of *S. aureus* is noteworthy, although, it may not provide significant benefits for diagnosis, as *S. aureus* typically causes acute, highly virulent infections [46]. However, in cases with low bacterial loads, such as in persistent or relapsing infections [47], saponin could be valuable. For *Cutibacteria*, which, alongside *S. epidermidis*, are major causes of difficult-to-diagnose PJIs [34,48], we observed an overall 1 log_10_ increase in CFU recovery. Considering the EBJIS PJI definition guidelines (i.e., ≥50 CFUs/mL) for sonication diagnosis, a 1 log_10_ increase in recovery could turn previously negative samples, which were considered contaminants due to low CFU counts, into positive PJI diagnoses. However, these guidelines should be carefully evaluated, as there are no standardized sonication protocols, particularly regarding the volume of sonication solution used [49].

We showed that vortexing with saponin at 0.001% (*w*/*v*) as compared to sonication demonstrated consistently higher bacterial recovery from implants made of various materials used in orthopedic surgeries, i.e., PE, TAV, CCM, and PMMA. This superiority was also reflected in the time-to-positivity (TTP), where saponin-treated samples showed markedly faster TTP. TTP is an important factor to consider in chronic infections. Although no studies have assessed the impact of reduced TTP on treatment duration or associated financial costs, it can be assumed that each additional day without a positive culture-delaying antibiotic resistance testing—may prolong treatment and potentially increase morbidity. This could also translate into prolonged hospital stays, resulting in increased healthcare costs. Therefore, reducing TTP could significantly improve patient outcomes and reduce costs.

We developed a simple 3D model mimicking a soft/connective-tissue PJI scenario, which more accurately reflects the clinical situation where bacterial localization often occurs at the implant-interface membrane [50,51]. Infection with *S. epidermidis* resulted in biofilm-like aggregates on the implant and in the tissue, similar to clinical PJI samples [52]. CFU recovery from homogenized tissue was significantly higher than with saline sonication, highlighting tissue as a potential infection reservoir. Identifying infected tissue intraoperatively is challenging, as bacteria are scattered around the implant. Biofilm on implants cannot be directly seen but can be cultured to pinpoint infection locations [53,54,55]. Treatment with saponin-vortexing yielded similar CFU recovery to homogenization, offering a quicker and cheaper alternative. While the model does not fully mimic thick tissue in the clinical PJI environment, it confirmed that vortexing with saponin solution improves CFU recovery compared to saline sonication. In comparison to in vitro models, the 3D model better represents the clinical PJI environment. Saponin’s effect on bacterial recovery was reduced in the 3D model, but even a 1-log increase in CFUs could have significant clinical implications.

As a proof-of-concept, we propose a novel two-component kit consisting of a saponin solution and a specialized transportation box for the efficient collection, transport, and processing of potentially infected implants (Appendix A). Once the implant is placed in the collection container, the container remains sealed. The lid of the container is equipped with a self-sealing access port that allows connecting to the squeeze bag in order to add the bacteria detachment solution to the implant. After processing (vortexing or sonication), a collection syringe can be attached to the container to withdraw the solution containing the detached bacteria. Following this step, routine diagnostic procedures can be followed Appendix A. This kit could enable more efficient and sterile processing of implants compared to current practices. Most laboratories use solid boxes that require multiple openings by different personnel, increasing the risk of contamination with skin bacteria. Using bags for collection also raises contamination risks [9]. Our two-component system eliminates the need to open the box once the implant is placed inside. Additionally, employing such a system would establish a clear protocol, particularly regarding the volume of solution to be added, as it is defined by the volume of the squeeze bag. This could standardize PJI infection definitions and make results across multiple centers more comparable. Furthermore, any solution other than saponin could be used in the squeeze bag to increase the chances of bacterial recovery. This box is currently being developed to marketability and produced by a service provider, and a patent has been filed for the design.

Our preliminary study has several limitations. First, we focused solely on saponin; other surfactants like Tween 20 or 80 may yield similar results. Second, we did not assess different biofilm incubation timepoints, as biofilm maturity might affect susceptibility to saponin. However, replicating real biofilm maturity in vitro is challenging, so we chose a timepoint that reflects CFU levels in clinically infected implants. Third, we tested saponin on a limited number of strains; given saponin’s soap-like mechanism, we do not anticipate strain resistance. Lastly, while promising, saponin should undergo clinical studies to validate its effectiveness and handling in a clinical setting. Thetransportation box is currently under development for future use in PJI diagnosis. Once fully developed and certified to use, we anticipate performing a clinical study similar to the one by Drago et al. [20], i.e., comparing sensitivity of our two-component kit including saponin versus standard saline sonication and tissue culture on large numbers of orthopedic implants from patients with assumed PJI.

Our findings show that vortexing with the biosurfactant saponin as compared to standard sonication improves bacterial recovery from infected orthopedic materials in vitro. We also proposed a novel collection and transportation system for more efficient processing of potentially infected implants. Pending clinical validation, these approaches could enhance PJI diagnosis, improve treatment outcomes, and reduce associated costs.

## Figures and Tables

**Figure 1 microorganisms-13-00836-f001:**
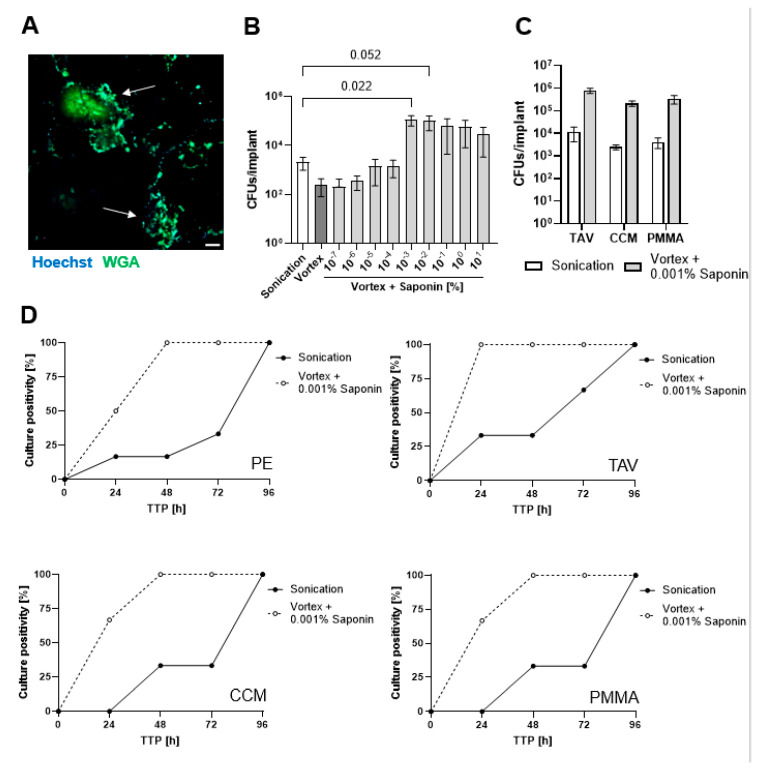
Saponin increases *S. epidermidis* CFU recovery from different orthopedic material. (**A**) CLSM image showing *S. epidermidis* biofilm aggregates (arrows) formed on the PE disc after 3 days of incubation in a physiological medium. The scale bar indicates 20 µm. (**B**) Enumeration of recovered *S. epidermidis* (CFUs/mL) from infected PE discs for the indicated treatment. Sonication was performed using PBS alone, whereas vortex treatment was conducted with PBS containing 0–10% (*w*/*v*) saponin. Statistical significance (*p*-value) was determined by One-Way ANOVA and is indicated above by the brackets for the compared groups. (**C**) Enumeration of recovered *S. epidermidis* (CFUs/mL) from infected TAV, CCM, and PMMA discs. Sonication was performed with PBS alone, while vortex treatment was conducted with 0.001% (*w*/*v*) saponin. Despite remarkable differences, none of the groups showed any statistical significance with Two-Way ANOVA. (**D**) Time-to-positivity (TTP) of diluted cultures from *S. epidermidis* infected PE, TAV, CCM, and PMMA discs. Sonication was performed with PBS alone, while vortexing was completed with PBS containing 0.001% (*w*/*v*) saponin, *n* = 3 biological replicates. Abbreviations: CFU, colony-forming units; CLSM, confocal laser scanning microscopy; PE, polyethylene; PBS, phosphate-buffered saline; TAV, titanium alloy Ti-6AI-4V; CCM, cobalt-chromium-molybdenum; PMMA, polymethyl methacrylate (PMMA)-based bone cement.

**Figure 2 microorganisms-13-00836-f002:**
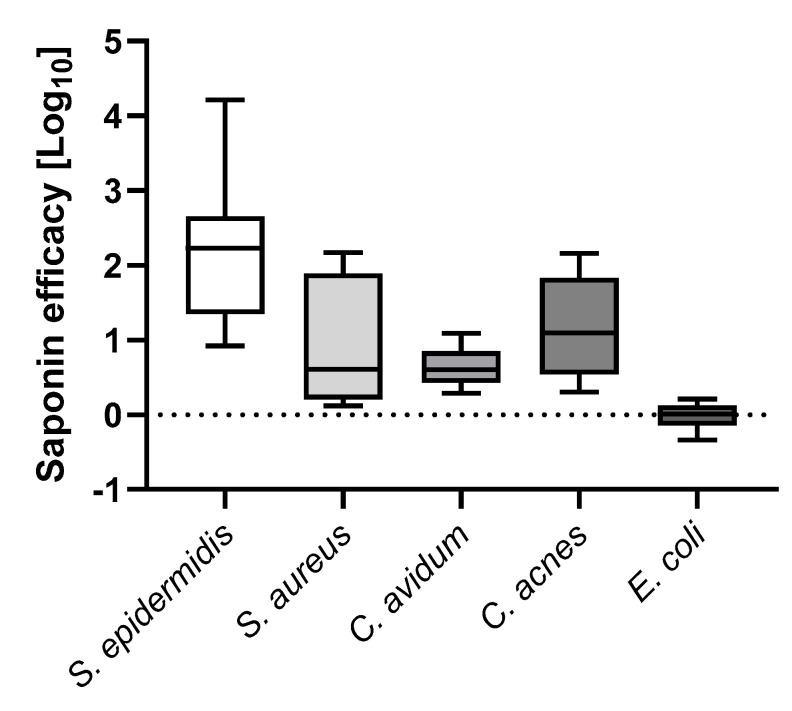
Efficacy of saponin-vortexing for different PJI-causing bacteria. Indicated are Log_10_ increased recovery of CFUs from infected PE discs of saponin (0.001% *w*/*v*) vortexing relative to saline sonication. The median values for the following clinical isolates are shown: *S. epidermidis*, *n* = 3 different clinical isolates; *S. aureus*, *n* = 2 different clinical isolates; *C. avidum*, *n* = 2 different clinical isolates; *C. acnes*, *n* = 2 different clinical isolates; *E. coli*, *n* = 2 different clinical isolates, *n* = 6–9 biological replicates per bacterial species. Abbreviations: CFU, colony-forming units.

**Figure 3 microorganisms-13-00836-f003:**
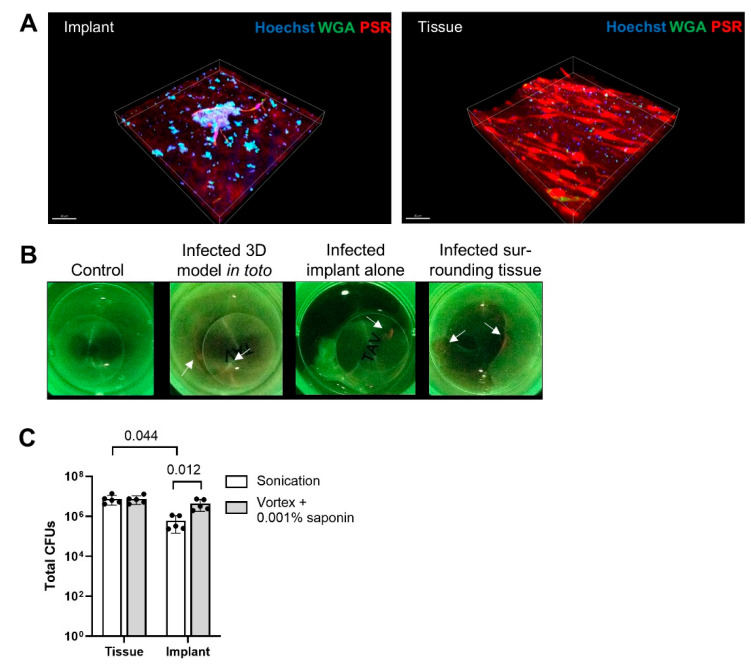
Novel 3D PJI Soft Tissue Model. (**A**) CLSM images demonstrating the presence of *S. epidermidis* biofilm aggregates (turquoise) on the TAV discs, surrounded by human cells and host tissue (red, collagen) and single *S. epidermidis* colonies (green) within the host tissue matrix colonized by human fibroblasts (red). Scale bars indicate 50 µm. (**B**) Fluorescent images obtained from the tissue model with the Moleculight i:*X*^TM^ device showing the host tissue (green, collagen) surrounding the 1 cm TAV discs. Red signal indicates *S. epidermidis* on the implant and in the tissue (arrows). (**C**) *S. epidermidis* CFU recovery of infected tissue models. The implant disc and tissue were collected and treated separately either with 0.001% (*w*/*v*) saponin in PBS combined with vortexing or saline sonication, *n* = 5 replicates. Abbreviations: CFUs, colony-forming units; CLSM, confocal laser scanning microscopy; TAV, titanium alloy Ti-6AI-4V.

## Data Availability

The original contributions presented in this study are included in the article/Appendix A. Further inquiries can be directed to the corresponding author.

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
