# Peer review of "Saponin Improves Recovery of Bacteria from Orthopaedic Implants for Enhanced Diagnosis Ex Vivo"

_microorganisms, 2025, doi:10.3390/microorganisms13040836_

Round 1
Reviewer 1 Report
Comments and Suggestions for Authors
The Authors investigated the use of saponin to enhance bacterial recovery from biofilm grown on orthopedic material in vitro.
The topic is interesting.
Introduction: please detail about additional chemical methods (eg DTT) doi: 10.1007/s11999.0000000000000060
Please define the rational of using saponin
Methods should be described more in details. It is not clear how samples were processed. How could both saponin and sonication be applied to the same sample? Also, which is the rational of combining both?
Any difference in performances among different pathogens?
Discussion: please compare these performances with other methods in the Literature (eg sonication and DTT)
Author Response
Please see the attachement. Line numbers refer to the version with track changes. We have uploaded a clean version as a Word file and a PDF file with track changes.

Reviewer 2 Report
Comments and Suggestions for Authors
The article addresses the important issue of peri-implant infection (PJI) diagnosis by evaluating the efficacy of saponin in improving bacterial recovery from biofilms on orthopedic implants. The paper provides valuable data on the potential use of biosurfactants as a complement or alternative to sonication. In addition, the authors present an innovative two-component transport system, which may have relevance in clinical practice. The paper is interesting, written generally correctly. Its structure corresponds to anuk procom. It has a few shortcomings that need to be corrected and completed before further proceedings and possible publication. I have provided detailed notes and comments on the work below.
Minor comments:
Although the authors conducted tests on the efficacy of saponin, they did not sufficiently provide justification for choosing the concentration of 0.001% w/w as optimal. To improve the quality of the paper, more detailed justification for this choice should be provided, as well as data on possible toxic effects on both bacteria and host cells.
The paper focuses on selected clinical isolates of S. epidermidis, S. aureus, C. avidum, C. acnes, C. granulosum, and E. coli, but lacks analysis of other relevant pathogens such as streptococci (Streptococcus spp.) or other Gram-negative bacteria. The authors should expand the analysis to include additional bacterial species, especially those frequently found in PJI.
The article compares the effectiveness of saponin to standard sonication, but omits a comparison with other substances used to remove biofilms, such as dithiothreitol (DTT) or enzymes. I feel it would be worthwhile to include additional comparative experiments or discuss the superiority of saponin over other methods in the discussion.
The authors should include other factors such as changes in the properties of bone cements can significantly affect the risk of peri-implant joint infection (PJI). The addition of antibiotics (ALBC) reduces this risk by providing a high concentration of the drug locally, although it can lead to the development of resistant strains. Increased cement porosity facilitates antibiotic diffusion but promotes biofilm formation, while lower porosity reduces bacterial colonization at the expense of poorer drug release. Modern modifications, such as silver nanoparticles, improve antibacterial properties, but excess additives can weaken cement strength. Please consider this aspect along with the necessary literature. Authors will find useful inforamation the series of works Effect of various admixtures on selected mechanical properties of medium viscosity bone cements: Part 1 , 2 and 3, as well as works: Effect of physiological fluids contamination on selected mechanical properties of acrylate bone cement; Seasoning polymethyl methacrylate (PMMA) bone cements with incorrect mix ratio; Analysis of the properties of bone cement with respect to its manufacturing and typical service lifetime conditions;
The authors did not provide information on the number of repetitions for each series of experiments, which is crucial for assessing the reliability of the results. More information on the number of repetitions should be added and a statistical analysis of the stability of the results should be performed. Please describe this aspect in more detail.
The results are based solely on an in vitro model, which limits their translatability to clinical practice. The authors should discuss the potential limitations of the in vitro study and suggest directions for further research in the clinical setting.
Author Response
please see the attachement. Reviewer 1 and 2 comments together
Line numbers refer to the version with track changes. We have uploaded a clean version as a Word file and a PDF file with track changes

Round 2
Reviewer 1 Report
Comments and Suggestions for Authors
The Authors made good efforts. The paper was significantly ameliorated.